# In Vitro Digestion and Fermentation of Different Ethanol-Fractional Polysaccharides from *Dendrobium officinale*: Molecular Decomposition and Regulation on Gut Microbiota

**DOI:** 10.3390/foods13111675

**Published:** 2024-05-27

**Authors:** Lei Xu, Hua Zhu, Peng Chen, Zhenhao Li, Kai Yang, Peilong Sun, Fangting Gu, Jianyong Wu, Ming Cai

**Affiliations:** 1Department of Food Science and Technology, Zhejiang University of Technology, Hangzhou 310014, China; xxxxl534331@163.com (L.X.); nmdg1zh@163.com (H.Z.); 18757073277@163.com (P.C.); yangkai@zjut.edu.cn (K.Y.); sun_pl@zjut.edu.cn (P.S.); 2Key Laboratory of Food Macromolecular Resources Processing Technology Research (Zhejiang University of Technology), China National Light Industry, Hangzhou 310014, China; 3Longevity Valley Botanical Co., Ltd., Jinhua 321200, China; zhenhao6@126.com; 4Department of Food Science & Nutrition, The Hong Kong Polytechnic University, Hung Hom, Kowloon, Hong Kong, China

**Keywords:** *Dendrobium officinale*, polysaccharides, digestion, gut microbiota, 16S rRNA

## Abstract

Polysaccharides from *Dendrobium officinale* have garnered attention for their diverse and well-documented biological activities. In this study, we isolated three ethanol-fractionated polysaccharides from *Dendrobium officinale* (EPDO) and investigated their digestive properties and effects on gut microbiota regulation in vitro. The results indicated that after simulating digestion in saliva, gastric, and small intestinal fluids, three EPDOs, EPDO-40, EPDO-60 and EPDO-80, with molecular weights (*Mw*) of 442.6, 268.3 and 50.8 kDa, respectively, could reach the large intestine with a retention rate exceeding 95%. During in vitro fermentation, the EPDOs were broken down in a “melting” manner, resulting in a decrease in their *Mw*. EPDO-60 degraded more rapidly than EPDO-40, likely due to its moderate *Mw*. After 24 h, the total production of short-chain fatty acids (SCFAs) for EPDO-60 reached 51.2 ± 1.9 mmol/L, which was higher than that of EPDO-80. Additionally, there was an increase in the relative abundance of *Bacteroides*, which are capable of metabolizing polysaccharides. EPDO-60 also promoted the growth of specific microbiota, including *Prevotella* 9 and *Parabacteroides*, which could potentially benefit from these polysaccharides. Most notably, by comparing the gut microbiota produced by different fermentation carbon sources, we identified the eight most differential gut microbiota specialized in polysaccharide metabolism at the genus level. Functional prediction of these eight differential genera suggested roles in controlling replication and repair, regulating metabolism, and managing genetic information transmission. This provides a new reference for elucidating the specific mechanisms by which EPDOs influence the human body. These findings offer new evidence to explain how EPDOs differ in their digestive properties and contribute to the establishment of a healthy gut microbiota environment in the human body.

## 1. Introduction

*Dendrobium officinale* has been utilized as a functional food in China for over two thousand years [1]. Its primary roles include regulating metabolic balance within the body and protecting bodily organs [2,3]. The polysaccharides of *Dendrobium officinale* are biological macromolecules predominantly composed of glucose and mannose, and these play a crucial role in its functional properties. These properties encompass antioxidant activity, reduction of blood glucose and lipid levels, and immune regulation [4,5,6,7].

The effect of polysaccharides on the large intestine and gut microbiota has become attractive because of its meaningful role in demonstrating the mechanisms of polysaccharides in treatment of the human body [8]. Lin et al. demonstrated that polysaccharides from *Laminaria japonica* can alleviate gestational diabetes by modulating the gut microbiota in mice [9]. Similarly, Zhou et al. discovered that polysaccharides from *Dendrobium officinale* may indirectly influence the immune system by regulating the intestinal microbiota [10]. The impact of polysaccharides on gut microbiota is specifically evident in the changes they induce in the intestinal flora and environment [11,12]. Li et al. found that upon entering the large intestine, *Dendrobium officinale* polysaccharides exhibited a gradual decrease in molecular size and stimulated the production of mannose. They also promoted the generation of polysaccharide metabolites, such as short-chain fatty acids (SCFAs), and affected the structure and abundance of the gut microbiota [13]. Wu et al. discovered that after fermentation, the aggregates of *Tremella fuciformis* polysaccharides were disrupted, with partial breakage of glycosidic bonds, leading to a decrease in molecular weight (*Mw*) from 1.8556 × 10^6^ to 0.523 × 10^5^ Da. Furthermore, FT-IR analysis indicated that the degree of esterification was lower in the fermented samples compared to the unfermented ones. These fermented products were also found to promote the proliferation of intestinal probiotics [14]. Fu et al. reported that polysaccharides from *Dendrobium officinale* increased gene abundance in amino acid and fatty acid metabolic pathways [15]. The results of metabolites also showed higher amino acid and fatty acid metabolites, which suggested that *Dendrobium officinale* polysaccharides had potential as prebiotics and might improve gastrointestinal health [16]. In our previous study, three fractions of polysaccharides with different molecular weights were extracted from *Dendrobium officinale*, which had different properties in anti-fatigue and affected the abundance of gut microbiota [17]. However, relationships between the different structural polysaccharides and their fermentation characteristics have never been clearly demonstrated. It has been shown that *Mw* has important influence on fermentation differences. The polysaccharide of 60S branched glucomannan extracted from *Artemisia sphaerocephala* Krasch seeds had a stronger ability to produce acidic substances after fermentation than did the 60P branched xylose [18]. It was reported that *okra* polysaccharides with β-1,4-gal*p* as the branch chain showed the ability to promote the growth of probiotics, highlighting the important role of β-configuration polysaccharides in biological activity [19]. However, different structural properties of *Dendrobium officinale* polysaccharides in digestion and fermentation have never been studied.

However, conventional studies often demonstrate poor efficacy, as they tend to only consider the abundance and diversity of a few microbial groups that have been the focus of research. Li et al. found a correlation between the abundance of polysaccharides and certain bacteria, but the direct influence of these microbial groups on the functionality of the gut microbiota remains uncertain [13]. These studies did not delve into a deeper classification of the gut microbiota and often overlooked differences in the abundance of polysaccharide-specialized microbial groups, thus missing out on functional analyses of the benefits derived from these gut microbial communities. Milani et al. conducted a systematic analysis of the gut microbiota based on functional characteristics, facilitating its organized classification [20]. *Bacteroides* to *Firmicutes* act as key players in carbohydrate degradation, carrying out metabolic functions in the gut. *Bifidobacterium* and *Faecalibacterium*, as gut microbial groups that utilize oligosaccharides to produce SCFAs, provide a continuous source of SCFAs through fermentation, benefiting probiotic bacteria. Most importantly, there are gut microbial groups that benefit from SCFA stimulation, and these groups serve as determinants of gut microbiota functionality. Lee et al. reviewed polysaccharide-specialized functional clusters through differential analysis of carbohydrates from different sources [21]. Therefore, the key to identifying functional clusters influenced by polysaccharides lies in comparative analysis of different types, making it crucial for us to conduct screening in order to uncover the gut microbial communities that truly exert the functional effects of polysaccharides.

In this study, we conducted in vitro simulated fermentation of ethanol-fractional polysaccharides from *Dendrobium officinale* (EPDO) to determine the changes in molecular weight and production of SCFAs during the digestive fermentation process. This revealed the transformations that occur during polysaccharide fermentation. Additionally, we performed a comparative analysis of the gut microbiota after simulated fermentation. Functional predictions were made for eight selected representative genera that exhibited significant polysaccharide characteristics. This will provide valuable insights for future research on the functional role of EPDO in the human body.

## 2. Materials and Methods

### 2.1. Materials

#### 2.1.1. Chemical Reagents

The reagents used in the digestion and fermentation experiments are listed in Appendix A. Additionally, phosphate buffer saline (PBS, 0.05 mol/L, pH = 6.8) and HCl were obtained from Yuan Ye Bio-Technology Co., Ltd. (Shanghai, China). Phenol, concentrated sulfuric acid, 3,5-dinitrosalicylic acid (DNS), trifluoroacetic acid (TFA), 1-phenyl-3-methyl-5-pyrazolone (PMP), chloroform, 2-ethylbutyric acid, acetic acid, propionic acid, and n-butyric acid were purchased from Aladdin Biochemical Technology Co., Ltd. (Shanghai, China). Standards of D-glucose and D-mannose were obtained from Sigma Aldrich Trading Co., Ltd. (Shanghai, China). TIANamp stool RNA kit was purchased from Tiangen Bio-Technology Co., Ltd. (Beijing, China).

#### 2.1.2. Preparation of Polysaccharides

EPDOs used in this study were prepared as our previous studies [17]. EPDOs were identified as 3 kinds of water-soluble 1,4-Man (*p*), named EPDO-40, -60 and -80, with different *Mw* about 452.5 kDa, 277.3 kDa and 53.6 kDa, carbohydrate contents of 84.2 ± 5.6%, 92.6 ± 3.3% and 88.7 ± 5.4%, and protein contents of 1.7 ± 0.3%, 1.6 ± 0.3% and 1.8 ± 0.2%, respectively.

#### 2.1.3. Fecal Samples

Based on the reported method [22] with modifications, our experimental protocol was reviewed and approved by the Ethics Committee of Zhejiang University of Technology (Approval No. MGS20220617147). Three healthy volunteers, aged 20–35 years, with a regular diet and no antibiotics within 6 months, were employed as subjects. On the day of experiment, the feces of three volunteers were collected with Anaeropack (Mitsubishi, Tokyo, Japan) to provide an anaerobic environment. Subsequently, the feces operated were diluted (1:10, *w*/*v*) by sufficient PBS in an anaerobic workstation (Concept 400, Baker Ruskinn, Königswinter, Germany). To eliminate the interindividual difference, the fecal inoculum was prepared after mixing the feces–PBS solution in equal volume from every volunteer.

### 2.2. Digestion In Vitro

Simulated digestive fluids of saliva and gastric and intestinal fluids were prepared according to the reported methods [10]. The saliva digestive fluid was mixed with powders of EPDO-40, -60 and -80 to reach a final concentration of about 8 mg/mL, respectively, using a platform shaker (Innova 2300, New Brunswick Scientific, Edison, NJ, USA) with a rotation speed of 280 rpm at 37 °C. Samples of 1 mL each were collected at 0, 5, 20 min during digestion and immediately put into a boiling water bath for 10 min and cooled to room temperature. After being centrifuged at 12,000 rpm for 10 min, the supernatant was immediately stored at −80 °C for subsequent study. An equal volume of deionized water was used as the control, and 3 parallel sets of tubes were set for each trial.

The prepared gastric and intestinal digestive fluid were mixed with EPDOs from the above digestion stage. After mixing, digestion samples were collected at 0, 2, 4 and 6 h for subsequent analysis. The other operations were the same as in the saliva simulated digestion.

### 2.3. Fermentation In Vitro

The process of fermentation in vitro is shown in Figure 1, according to the reported method [9]. A 1 mL quantity of fecal inoculum, 4 mL large intestinal simulated fermentation fluid, and 40 mg EPDOs were mixed. Each sample was divided into four groups for 0, 6, 12, and 24 h, fermented at 37 °C. All samples were collected for analysis or kept at −80 °C for subsequent experiments. Oligofructose (FOS) of 8 mg/mL was set as the positive group, and carbon source as the blank.

### 2.4. Determination of Carbohydrate and Reducing Sugar Contents

Carbohydrate content of sample was determined by the phenol-sulfuric acid method with D-glucose as standard [23]. The sample reacted with phenol sulfuric acid in a boiling water bath, and then the absorbance was measured by microplate reader (Multiskan MK3, Thermo Fisher Scientific, Waltham, MA, USA) at 490 nm.

Reducing sugar content of sample was determined by DNS method with slight modification, with D-glucose as standard [24]. The sample reacted with DNS in a boiling water bath, and then the absorbance was measured by microplate reader at 540 nm.

According to Hu & Di’s report [25], the digestibility of EPDOs was calculated as a ratio of the content of reducing sugar at time *t* (CR_t_) to the content of carbohydrate at start (C_to_), as shown in Equation (1). The degradation rate in fermentation was calculated as a ratio of the content of carbohydrate at time *t* (C_t_) to the content of carbohydrate at start (C_to_), as shown in Equation (2).
(1)$$\mathrm{Digestibility}\left(\%\right)=\frac{{\mathrm{CR}}_{t}}{C_{\mathrm{to}}}\times 100\%$$
(2)$$\mathrm{Degradation}\mathrm{rate}\left(\%\right)=\frac{C_{t}}{C_{\mathrm{to}}}\times 100\%$$

### 2.5. Determination of Molecule Weights and Monosaccharides

Molecule weights were determined according to the reported method with slight modification [26]. A 100 μL sample was injected into a Breeze 2 HPLC system (Waters 1515, Waters Corporation, Milford, MA, USA) combined with an RI-detector (Waters 2414, Waters Corporation, Milford, MA, USA), with a Waters Ultrahydrogel 500 column.

Monosaccharides of polysaccharides were determined according to the reported HPLC method with slight modification [27]. The 100 μL sample was processed on the HPLC system after PMP derivatization, with ZORBAX Eclipes XDB-C18 (Agilent Technologies, Santa Clara, CA, USA) at 40 °C. The mobile phase was MiliQ water: acetonitrile = 82:18 at 0.5 mL/min, with a standard of D-glucose and D-mannose.

### 2.6. Determination of pH

The pH was determined according to the reported method [10]. After fermentation for 0, 6, 12, and 24 h, the sample was measured by a pH meter (PHS-3C, Sanxin, Shenyang, China) at room temperature.

### 2.7. Determination of SCFAs

SCFA determination was carried out according to the reported method [28] with slight modification. The samples mixed with 10 μL 2-ethylbutyric acid were collected after fermentation and measured by gas chromatography (Agilent 6890N, Agilent Technologies, Santa Clara, CA, USA) with HP-INNOWAX column and FID detector. The carrier gas was nitrogen at a flow rate of 19.0 mL/min and shunt ratio of 1:10. The flow rates of air and hydrogen were set at 300 and 30 mL/min, respectively. The heating program was 100–180 °C at a speed of 4 °C/min and run for 20.5 min.

### 2.8. Sequence of Gut Microbiota

#### 2.8.1. DNA Extraction and Microbiota Sequencing

The microbiota was determined by the 16s rRNA method with modification [29]. After fermentation for 0 and 24 h, RNA of the sample was extracted by Tianamp stool RNA kit, conducting the operation as described in the kit manuscripts. After extraction, 16s rRNA sequencing was conducted by Mingke Biotechnology Co., Ltd. (Hangzhou, China). Three random samples from the BLANK, EPDO-40, -60, -80 and FOS groups that were fermented for 24 h participated in 16s sequencing, where 1, 2, and 3 represent three parallel samples, respectively, while BLANK-0 was the unfermented sample (*t* = 0).

For 16s sequence, the V3-V4 region of the bacterial 16S ribosomal RNA genes were amplified using barcoded primers: forward primer (5′-3′): CCTACGGGRSGCAGCAG (341 F) and reverse primer (5′-3′): GGACTACVVGGGTATCTAATC (806 R), with a denaturation step at 95 °C for 3 min, followed by 30 cycles at 98 °C for 20 s, 58 °C for 15 s, and 72 °C for 20 s and a final extension at 72 °C for 5 min. All quantified amplicons were pooled to equalize concentrations for sequencing using Illumina MiSeq (Illumina, Inc., San Diego, CA, USA).

#### 2.8.2. Bioinformatic Analysis

α-diversity indices, including the observed species, abundance-based coverage estimator (ACE), Chao1 estimator, Shannon, Simpson, Evenness and PD whole tree indices, were calculated at a 97% similarity level.

β-diversity was measured by unweighted UniFrac, weighted UniFrac, Jaccard, and Bray. Curtis distances were calculated by QIlME2, which were visualized by principal coordinate analysis (PCoA).

The differences in the composition of the gut microbiota at different taxonomic levels were analyzed with the Statistical Analysis of Metagenomic Profiles (STAMP) software package v2.1.3 and the linear discriminant analysis (LDA) effect size (LEfSe) method.

The functional prediction was analyzed using phylogenetic investigation of communities by reconstruction of unobserved states to identify enrichment of Kyoto Encyclopedia of Genes and Genomes (KEGG) pathways [30].

#### 2.8.3. Filtration of Gut Microbiota

The screening protocol was developed based on previous studies and findings from our lab [16,21]. The gut microbiota, after fermentation of other polysaccharides, starch, and oligosaccharides, were separately included as a contrast of carbohydrate metabolism. The gut microbiota after flavone and phenolic acid fermentation were included as contrast of SCFA production. After the LEfse analysis of these different categories of carbon sources were included, there were significant differences in the unique functional flora produced after each fermentation.

The intestinal flora sample using EPDO-40 as the fermentation product was classified into the polysaccharide (APS) category during the analysis, abbreviated as P40 and parallel to 3 samples, and the data source was this study. Other carbon sources and classifications are shown in Appendix A.

#### 2.8.4. Bioinformatic Data Accession

The 16s rRNA data from this study were deposited in the GenBank Sequence Read Archive with accession number PRJNA1097923.

### 2.9. Statistical Analysis

The experiments were performed in triplicate. The data are presented as the average ± standard deviation (SD). EPDO modification, pH value, and SCFA production data were analyzed using one-way analysis of variance (ANOVA) and compared using the Tukey test with a confidence level of 95% using SPSS software (version 22.0, SPSS, Chicago, IL, USA). R (version 4.3.1) and R packages were used for preparation of graphs. Statistical significance was determined at a *p*-value of less than 0.05.

## 3. Results

### 3.1. Retention of EPDOs after Digestion In Vitro

The properties of EPDOs’ *Mw* in saliva, gastric, and small intestine digestion are shown as HPGPC profiles in Figure 2. Compared to the original samples, the *Mw* of EPDOs after digestion had slight reductions. However, these reductions were not significant (*p* > 0.05), and the changes of *Mw*, carbohydrate contents, and reducing sugars are shown in Table 1.

It was indicated that in the saliva digestion, the digestibility of these three EPDOs were all less than 5%. However, three EPDOs showed different degrees of degradation in gastric digestion. EPDO-40 showed an increase in reducing sugar content from 2% to 3%. The digestibility of EPDO-60 and EPDO-80 remained almost stable in gastric digestion. Conversely, the reducing sugar content increased in the intestinal digestion. However, the changes in carbohydrate contents of these three EPDOs after digestion were not significant (*p* > 0.05).

### 3.2. Degradation of EPDOs after Fermentation In Vitro

Carbohydrate content of samples gradually decreased, and the degradation rate gradually increased during fermentation, as well as the reducing sugar content. This indicates that the degradation rates of EPDO-40 and EPDO-80 became slower during fermentation, as shown in Figure 3 and Table 2. This was inconsistent with the phenomena of EPDO-60 and FOS, but they had larger and continued high degradation rates to be 63.8% and 70.4%, respectively, after 24 h. The ratio of mannose to glucose was close to the original, about 2.6:1, 4.0:1 and 2.0:1 for EPDO-40, EPDO-60 and EPDO-80, respectively.

The changes to the EPDOs before and after fermentation are shown in Figure 4. At first, the peak of polysaccharides appeared at 18–40 min. The main changes in the fermentation process were that the retention time of these polysaccharides moved backward and the peak gradually decreased. After 24 h of fermentation, these peaks basically disappeared, while the area of peak after 40 min increased. This might have been caused by the continuous reduction of polysaccharides, resulting in small molecules such as monosaccharides and oligosaccharides being produced.

### 3.3. Effect of EPDOs on Intestinal Environment

As shown in Figure 5, the pH of the fermentation solution of these three EPDOs decreased during fermentation. Compared to that of the FOS group, the acidity of the EPDOs decreased slowly, and pH changed less during the last 12 h.

In Figure 6, the SCFAs in each group gradually increased. FOS had an extremely strong promotion effect on acetic acid secretion, while EPDO-80 facilitated the production of propionic acid. After 24 h, levels of n-butyric acid in the EPDO-60 fermentation solution were significantly higher than in other groups. In general, the acid production capacity of EPDO-60 was almost comparable to that of FOS, and it showed no reduction after 12 h.

### 3.4. Effect of EPDOs on Gut Microbiota

As shown in Figure 7, the gut microbiota of each group had good consistency from the hierarchical cluster analysis, and clear separation of gut microbiota could be discovered among different groups. This indicates that the effect of the BLANK-0 group was the most distinct from that of the BLANK-1, -2, and -3 groups, but similar to that of the EPDO-60 group.

The diversity of the communities was calculated by measuring Shannon’s index as shown in Figure 8A, which accounts for both species’ richness and evenness. This indicates that gut microbiota in both EPDOs and FOS groups showed a trend of decline, while the species of gut microbiota increased significantly when no carbon source was given.

Figure 8B shows the accumulation of gut microbiota at the phylum level, mainly considering the ratio of *Bacteroides* to *Firmicutes* (B/F). In the BLANK group, the B/F value decreased. Conversely, it increased after being given EPDOs, of which the EPDO-60 group was the largest. The change of B/F in the FOS group was insignificant, as listed in Appendix A.

An accumulation gut microbiota at the genus level is shown in Figure 8C, which mainly considered the comparison of gut microbiota abundance in different groups under specific subdivisions. It was focused on the microbiota, especially *Eschcrichia-Shigella*, *Proteobact* and *Megamonas* in the EPDO-60 group, and the abundance of these bacteria decreased compared with that of the BLANK-0 group. On the other hand, the abundance of *Prevotella* 9, *Lachnospria,* and *Lachnoclostridiun* increased. FOS had a completely different effect on the abundance of microbiota at the genus level. The abundance of *Bifidobacterium* and *Lactobacillus* increased significantly, as listed in Appendix A.

Figure 8D shows the numerical manifestation of gut microbiota in each group after LDA analysis, which indicated the specific type of gut microbiota. In the EPDO-60 group, compared with 142 kinds of intestinal bacteria with SCFA-producing genes were listed in Appendix A, *Ruminiclostridium*, *Parabacteroides* and *Blautia* showed positive.

### 3.5. Filtration Polysaccharide-Specialized Gut Microbiota and Functional Analysis

Figure 9A shows the significant functional differences among groups at Level 2 after KEGG analysis. The BLANK group exhibited a relatively balanced functionality of the gut microbiota, including functions related to cellular processes, human diseases, and environmental information processing. The EPDO group primarily demonstrated functions related to polysaccharide degradation and metabolism. Many of the significantly different functions were categorized under Metabolism (Level 1). Particularly in the P60 group, almost all the significantly different functions were metabolism-related. The KEGG Level classification is provided in Appendix A. These results did not reflect the advantages of EPDO in shaping the gut microbiota of the organisms. Instead, some dominant polysaccharide-specialized bacteria occupied the functional advantages, hindering the analysis of the specific functional characteristics of polysaccharides.

Therefore, it was crucial to screen the gut microbiota. The differential analysis of the gut microbiota after fermentation with other polysaccharides, starch, oligosaccharides, polyphenols, and flavonoids is shown in Appendix A. A total of 25 significantly different gut microbial groups were selected for polysaccharides. Due to the limitations of the analysis capacity, this study temporarily selected the top 8 significantly different genera as the uniquely advantageous bacteria for polysaccharides. These genera were *Bacteroides*, *Dialister*, *Flavonifractor*, *Holdemania*, *Sellimonas*, *Lachnospira*, *Parasutterella*, and *Fusicatenibacter*. These eight genera of gut microbial communities showed good classification attributes in the gut microbiota compared to other non-APS samples, with AUC values > 50%, as shown in Appendix A. These eight genera exhibited differences in abundance among different samples, as demonstrated in Figure 9A. *Bacteroides*, due to its excellent ability to degrade polysaccharides, exhibited dominant abundance in the APS group. The other seven genera seemed to display specific functionality, with their abundance being slightly higher than that of the non-APS group. Among them, *Lachnospira* and *Fusicatenibacter* also exhibited different abundances in the P40, P60, and P80 groups. However, discussing the functionality of individual gut microbial communities at this point was not meaningful because these bacteria, as a combination, had different abundances in different groups and performed different gut functions.

After filtering, KEGG analysis was performed on the eight genera based on their differential abundance in the P40, P60, and P80 groups. It was found that the BLANK group exhibited predominantly metabolism-related functions, while the P40 group showed significant differences in genetic information processing. The P60 group not only had differences in metabolism-related functions but also exhibited membrane transport and cellular processes. The P80 group showed replication and repair. Figure 9B demonstrated significant differences compared to the previous unfiltered KEGG analysis.

## 4. Discussion

### 4.1. EPDOs Achieve into the Large Intestine Steadily

According to the results, the answer was that all three samples of EPDOs could enter into the large intestine with a retention rate above 95%, which was similar to the results of Brodkorb’s report [31], which stated that digestion had no significant effect on polysaccharides. In addition, *Dendrobium officinale* polysaccharides could be retained in the large intestine due to lacking CAZymes at this stage [9]. Usually, the decrease of *Mw* and the production of reducing sugars happened at the gastric digestion stage. By comparison, the resulting hydrolysis of polysaccharides might be caused by the strong acidity in the gastric environment [28]. However, the amount of this hydrolysis was relatively small, and finally the EPDOs could work in the large intestine with a high retention rate.

### 4.2. EPDOs Degrade in the Large Intestine in a “Melting” Manner

It was suggested that EPDOs were degraded in a “melting” manner, and the opposite was the “chewing” way. Citing the “candy model”, some people prefer to bite hard candy directly for greater sweetness and efficiency, and some like to slowly let it melt in their mouths, so as to achieve a longer sweetness effect.

According to our analysis, the former hypothesis was rejected because the *Mw* of EPDOs decreased after fermentation. However, the peak height of HPGPC curve did not change, which meant the molar amount of EPDOs might not be changed [32]. It was obvious that the proportion of degraded monosaccharides and original sample were approximately consistent [13]. It was speculated that EPDO-60 with a high content of mannose and glucose exhibited a relative stronger degradation activity by binding to receptors on the gut microbiota surface with high affinity [9,16,33]. Moreover, EPDO-60 showed a higher degradation rate, indicating that EPDO-60 was favored in fermentation.

The main system for polysaccharide degradation in *Firmicutes* might be the ATP-binding cassette transport system [34,35]. The carbohydrate recognition proteins on the cell surface would recognize monosaccharides exposed to the outside of the molecule, 3–7 units in length, and then transport them into the cytoplasm for decomposition, with glucose and fructose as the main recognition site [36]. Therefore, more branched chains meant that there were more opportunities to be recognized by the binding sites on the cell surface of *Firmicutes* [37]. In our previous study, low molecular weight of EPDOs such as EPDO-80 with 53.6 kDa could be degraded easily. Its production of monosaccharides was slow because the proportion of branched chains was less than that of the others, in which the 3 linkage site monosaccharide residue contents are EPDO-40: 3.6%, EPDO-60: 2.1%, and EPDO-80: 1.0% [17]. This indicates that EPDO-80 had a low acid production rate and low SCFA production, which would merit further analysis.

### 4.3. Production of SCFA by EPDO in the Large Intestine

The most intuitive explanation from this study was that after EPDOs degraded into monosaccharides, SCFA was produced by gut microbiota fermentation, which formed a relatively healthy cycle [38]. Gut microbiota could make the EPDOs decompose and be utilized, and the SCFA production might have a better effect on the body [39,40]. For example, EPDO-80 was conducive to the formation of propionic acid, which is an important manifestation of a certain function [40]. EPDO-60 could produce a lot of n-butyric acid after being used by intestinal flora. The effect of these n-butyric acids on the body was reflected in improved immunity [41]. Moreover, the production of SCFA directly leads to a change of pH in the intestinal environment [10]. This change in the intestinal environment has a direct selection effect on the gut microbiota [42].

In addition, it was found that EPDO-60 and FOS groups could maintain a high growth rate in total SCFAs during fermentation, which was not affected by time. This might be directly related to the conclusions mentioned that carbon sources were preferred by gut microbiota because of their structure. An appropriate structure could ensure that EPDO-60 provided a steady stream of monosaccharides for fermentation when degradation happened, more SCFAs were produced [43]. Therefore, EPDO-60 had great potential in promoting the improvement of gut microbiota.

### 4.4. EPDOs Promoted the Growth of Gut Microbiota

In Figure 7, it can be found that under these various carbon source conditions, the gut microbiota had different development directions, suggesting that different structures of carbon sources led to different effects [44]. In the blank group, the gut microbiota would deviate from its original state. When EPDOs were used as a carbon source, they could effectively protect the original species and structure of gut microbiota. The reason for this is likely to be that after polysaccharides were degraded by gut microbiota, various metabolites produced made changes in the intestinal environment. In this environment, the original gut microbiota was more competitive, so there was a slight change [45].

As shown in Figure 8A, the growth of gut microbiota species was indeed constrained under the carbon sources, which was confirmed by the previous study [18]. However, whether it was caused by metabolites, especially SCFA, needs further analysis.

In Figure 8B, EPDOs could significantly lead to the proliferation of *Bacteroides*, and the increase of *Bacteroides* could promote the secretion of polysaccharide lyase and production of glycosidic bond hydrolase [17]. This was the inevitable result of carbon source and could further promote the degradation of polysaccharides. As a result, it could be found that polysaccharide was one of the main factors affecting the abundance and development direction of gut microbiota [29].

*Eschcrichia-Shigella*, *Proteobact* and *Megamonas* are considered to be adverse microbiota to intestinal health [20]. They have the adverse functions of toxin generation, cancer induction, and infection promotion, respectively [46]. In Figure 8C, the abundance of these gut microbiota decreased when EPDO-60 was used, which meant EPDO-60 might prevent intestinal cancer or infection. *Prevotella* 9, *Lachnospria* and *Lachnoclostridiun* family are considered as having protective effects on intestinal health [47,48]. They have the functions of regulating immunity, changing metabolism, anti-tumor and protecting intestinal tract after large intestine inflammation, respectively [49]. The increased abundance of these gut microbiota suggested that EPDO-60 might have immune-boosting properties. FOS is a recognized intestinal prebiotic, and *Bifidobacterium* and *Lactobacillus* were significantly increased after FOS treatment [34]. This also revealed that EPDOs and FOS have two different effect mechanisms. The differences in structure of carbon sources were likely to be the result of various gut microbiota environmental. Finally, it could be demonstrated that the main function of EPDO-60 was in the direction of immune regulation and anti-inflammatory.

The specific type of gut microbiota in EPDO-60 group is shown in Figure 8D. Compared with Zhao’s study [50], this indicated that SCFA producers, including *Ruminiclostridium*, *Parabacteroides* and *Blautia* became competitive in the EPDO-60 group.

### 4.5. The Gut Microbiota of Polysaccharide-Specialized and Their Functions

Figure 9A describes the results of KEGG functional analysis before filtering. Prior to the classification filtering, the gut microbiota of the EPDO group, after polysaccharide fermentation, did not exhibit significant functional advantages. Instead, the BLANK group showed more beneficial effects on the human body. This was because the intake of polysaccharides led to a significant increase in the abundance of *Bacteroides* and an increase in the diversity of carbohydrate-degrading bacterial species [21,22]. This phenomenon overshadowed the growth of beneficial bacteria induced by EPDO, as abundance is an important factor in KEGG functional analysis. Hu et al. also observed a similar phenomenon in their study on the gut microbiota after fermentation of a certain polysaccharide, with a significant increase in carbohydrate-degrading bacteria [25]. In addition, metabolism, carbohydrate metabolism, glycan biosynthesis and metabolism, lipid metabolism, biosynthesis of other secondary metabolites, galactose metabolism, other glycan degradation, and transport and catabolism were the primary phenotypic enrichment KEGG pathways in the P60 group, potentially related to its high total SCFA content [51]. According to Fang et al., non-starch polysaccharides can facilitate the metabolism of starch, sucrose, and oligosaccharides by boosting the production of short-chain fatty acids in the intestine [52]. However, the specific aspects of polysaccharide-specialized improvements to the gut microbiota were not clearly elucidated.

According to previous reports, the gut microbiota can be categorized at least into three main groups that perform carbohydrate metabolism, short-chain fatty acid production, and beneficial functions for the human body [15]. After filtering, the unique functions of various gut microbial communities were revealed, as shown in Figure 9B. Notably, the P80 group exhibited unique functions related to replication and repair. Previous studies by Lai et al. have shown that gut microbial communities with genera functional annotations related to replication and repair often have anti-inflammatory and tissue repair-promoting functions [53]. This finding suggests that EPDO-80 may have a beneficial direction towards promoting gastrointestinal protection. The P60 group, with gene annotations related to amino acid synthesis and metabolism, indicates a potential correlation between EPDO-60 and antioxidation and anti-aging effects in the body. In our previous study, EPDO-60 demonstrated significant benefits in alleviating physical fatigue in mice, helping them to rapidly eliminate metabolic waste after exhaustive swimming [17]. This could be attributed to the inseparable link with their gut microbiota. The gut microbiota of EPDO-40 after fermentation showed significant involvement in genetic information processing, suggesting a potential association with the growth and development of the organism. However, further experiments are needed to validate the specific underlying reasons.

To summarize, these results successfully made a reasonable connection between the polysaccharides molecule changes and regulation of gut microbiota. First, the supply of carbon sources led to SCFA-producing. Second, the produced SCFAs contributed to the acidic environment of the intestine, creating a suitable growth environment for some beneficial bacteria. Finally, beneficial bacteria contribute to the human body by promoting the stability of the intestine and playing the role of immune regulation, as shown in Figure 10.

## 5. Conclusions

This study demonstrated that after simulating saliva, gastric, and small intestinal digestions, three EPDOs were minimally degraded due to the absence of polysaccharide-decomposing enzymes in the digestive tract. Only a small fraction of EPDOs underwent degradation during gastric digestion, which was attributed to the acidic environment of the stomach. The EPDOs could reach the large intestine with a retention rate exceeding 95%. As a carbon source, EPDOs were broken down by the gut microbiota in a “melting” manner, leading to a significant reduction in molecular weight (*Mw*). Furthermore, there was an increase in the relative abundance of *Bacteroides*, which possess the capability to metabolize polysaccharides and are subsequently utilized by SCFA-producing bacteria. The SCFAs could alter the pH of the large intestine, influencing the abundance of gut microbiota. As functional bacteria unique to polysaccharides, eight gut microbiota exhibited distinct functional roles, ultimately aiding in the enhancement of the body’s immune system.

All these findings provide evidence to answer the question of how EPDOs exert their biological activities and offer important guidance for the future development of EPDO-based health foods. The main function of these foods will be to regulate the intestinal flora of the human body in the direction of health care products, or as carriers for drugs, warranting continued in-depth research and development.

## Figures and Tables

**Figure 1 foods-13-01675-f001:**
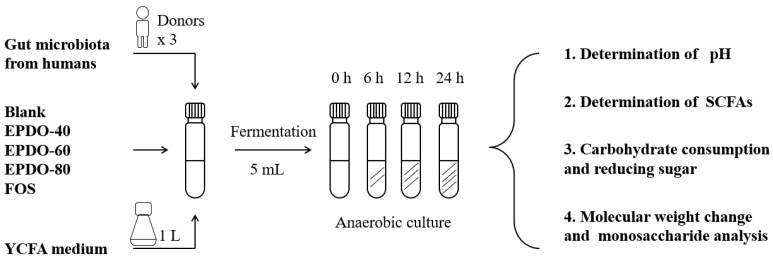
Process of EPDO fermentation by anaerobic culture.

**Figure 2 foods-13-01675-f002:**
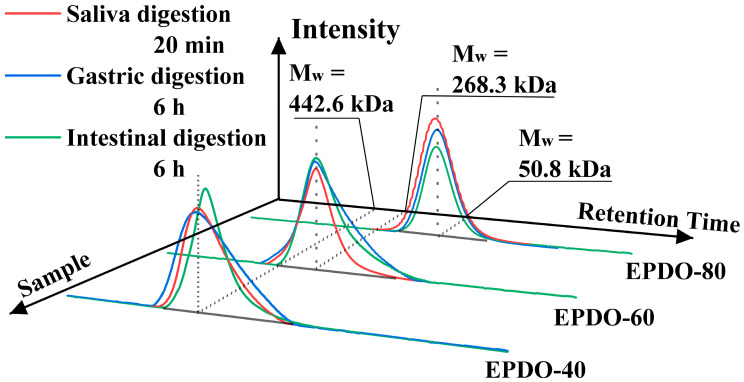
Changes of molecule weight of EPDOs after the digestion.

**Figure 3 foods-13-01675-f003:**
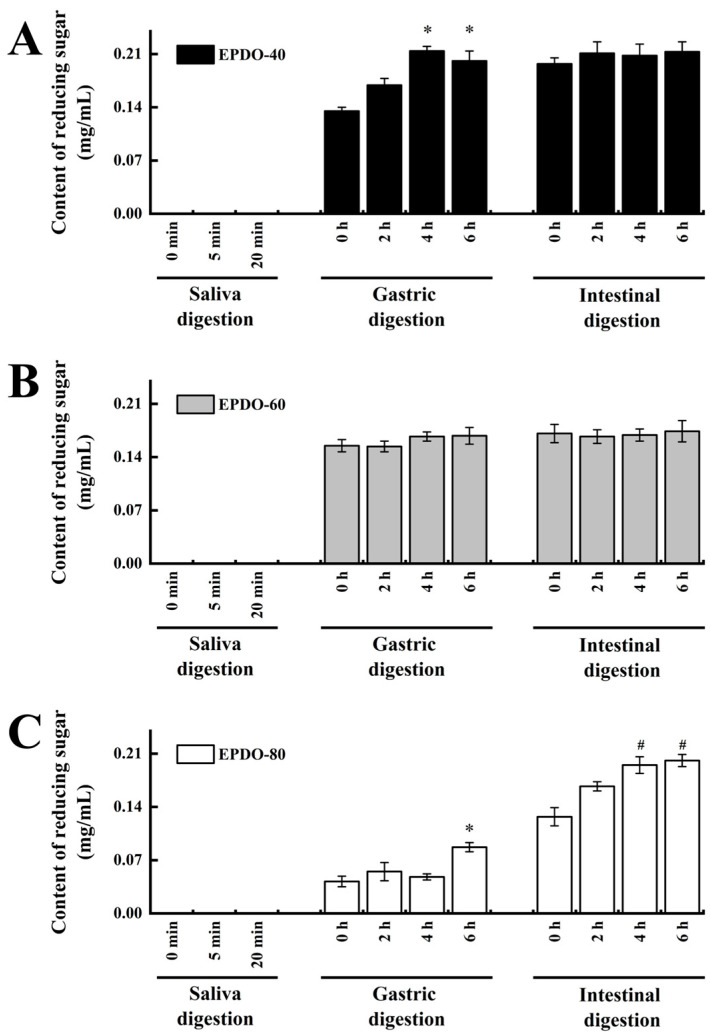
Reducing sugar content during the digestion stage. (**A**) EPDO-40; (**B**) EPDO-60; (**C**) EPDP-80. The value bars with signs denote statistically significant differences, * *p* < 0.05 in comparison with 0 h of gastric digestion; ^#^ *p* < 0.05 in comparison with 0 h of intestinal digestion.

**Figure 4 foods-13-01675-f004:**
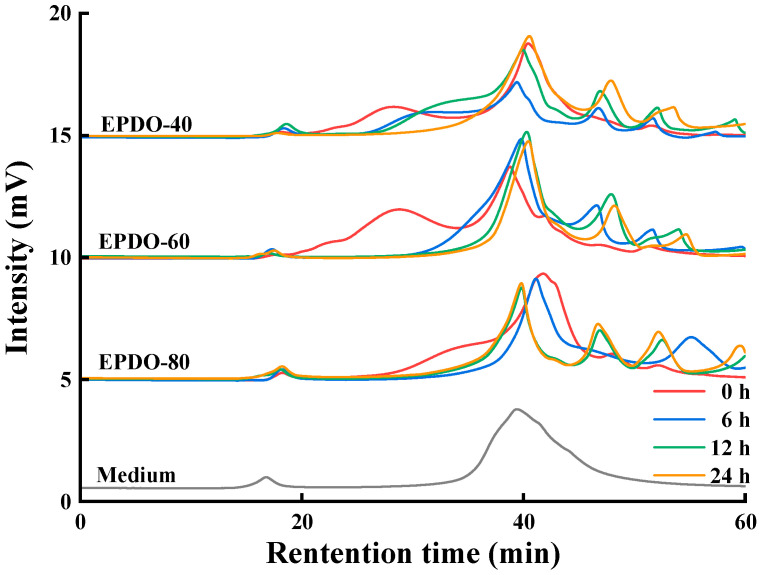
Changes to molecule weight of EPDOs after fermentation for 0, 6, 12, and 24 h.

**Figure 5 foods-13-01675-f005:**
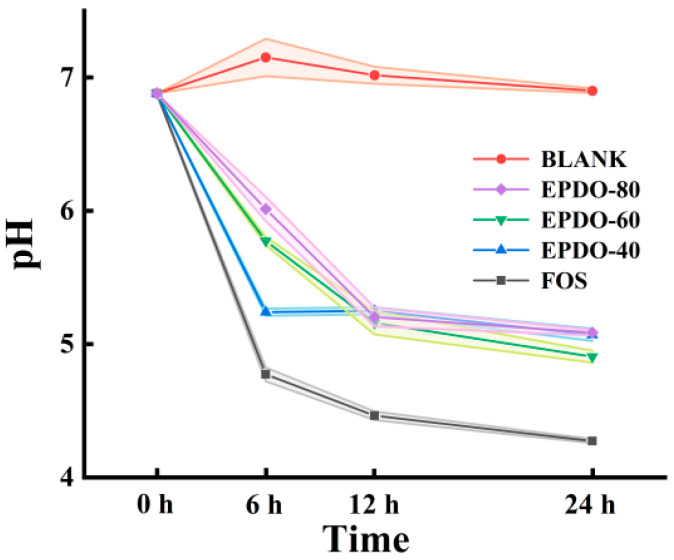
Changes in pH in the EPDO, FOS and blank groups from 0 to 24 h of fermentation.

**Figure 6 foods-13-01675-f006:**
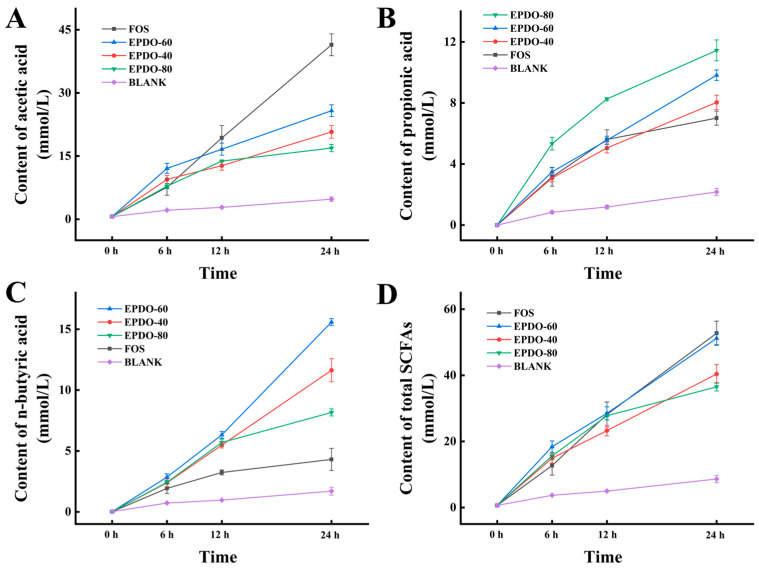
Production of SCFAs including the acetic acid (**A**), propionic acid (**B**), n-butyric acid (**C**), and total SCFAs (**D**) in EPDOs, FOS and blank groups from 0 to 24 h of fermentation.

**Figure 7 foods-13-01675-f007:**
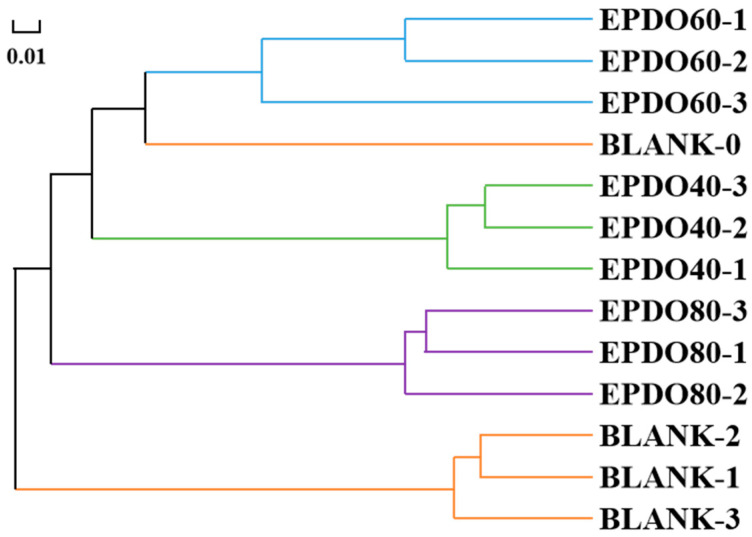
Correlation analysis of gut microbiota clusters.

**Figure 8 foods-13-01675-f008:**
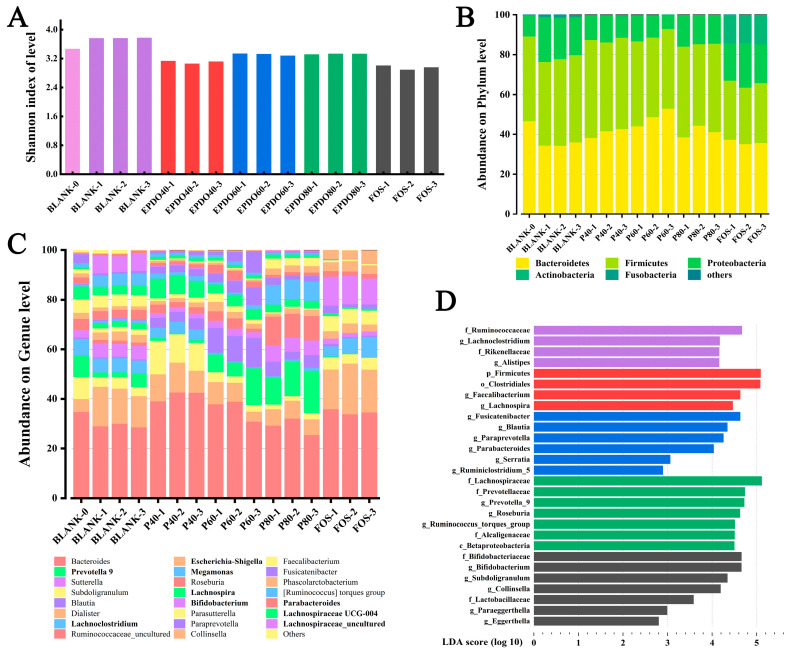
Analysis of gut microbiota α and β diversity: (**A**) Shannon index; (**B**) abundance on phylum level; (**C**) abundance on genus level; (**D**) uniqueness index between groups.

**Figure 9 foods-13-01675-f009:**
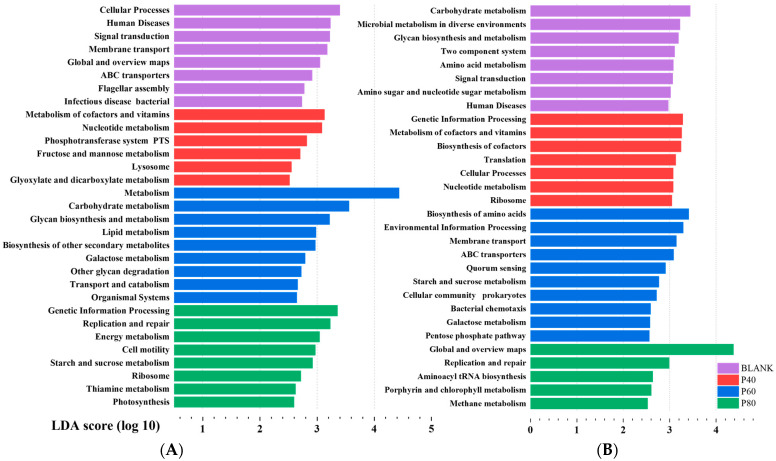
Lefse analysis of gut microbiota in KEGG functional analysis before (**A**) and after (**B**) filtration.

**Figure 10 foods-13-01675-f010:**
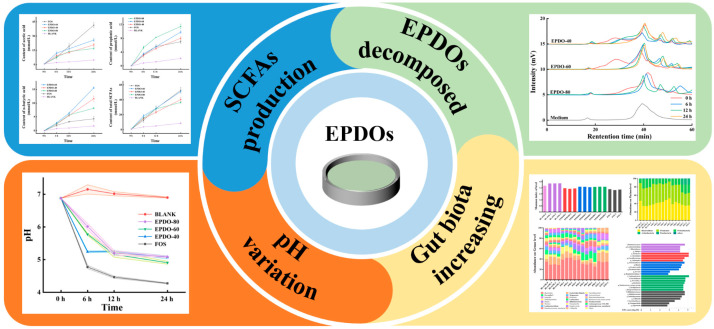
The mechanism of EPDO regulating gut microbiota by affecting intestinal environment.

**Table 1 foods-13-01675-t001:** EPDOs variation with time in the digestion.

Sample Name		Time	*Mw* (kDa)	Total Carbohydrate (mg/mL)	Reducing Sugar (mg/mL)	Digestibility (%)
Saliva	EPDO-40	0 min	452.5 ± 12.2	6.54 ± 0.02	ND ^a^	/ ^b^
digestion		5 min	444.1 ± 8.5	6.65 ± 0.05	ND	/
		20 min	449.4 ± 10.3	6.53 ± 0.04	ND	/
	EPDO-60	0 min	267.3 ± 7.2	7.26 ± 0.07	ND	/
		5 min	274.2 ± 8.5	7.21 ± 0.06	ND	/
		20 min	271.4 ± 5.3	7.18 ± 0.08	ND	/
	EPDO-80	0 min	53.6 ± 1.9	6.89 ± 0.03	ND	/
		5 min	54.1 ± 3.7	6.68 ± 0.08	ND	/
		20 min	52.1 ± 4.7	6.71 ± 0.01	ND	/
Gastric	EPDO-40	0 h	445.6 ± 11.5	6.64 ± 0.05	0.135 ± 0.005	2.0
digestion		2 h	442.6 ± 9.8	6.53 ± 0.03	0.169 ± 0.009	2.6
		4 h	441.1 ± 8.9	6.51 ± 0.05	0.214 ± 0.006	3.3
		6 h	440.5 ± 10.8	6.57 ± 0.03	0.201 ± 0.013	3.1
	EPDO-60	0 h	271.4 ± 8.1	7.31 ± 0.04	0.155 ± 0.008	2.1
		2 h	267.9 ± 7.5	7.25 ± 0.05	0.154 ± 0.007	2.1
		4 h	268.3 ± 7.9	7.21 ± 0.05	0.167 ± 0.006	2.3
		6 h	264.8 ± 8.6	7.26 ± 0.07	0.168 ± 0.011	2.3
	EPDO-80	0 h	53.5 ± 2.1	6.84 ± 0.05	0.042 ± 0.007	0.6
		2 h	51.1 ± 3.5	6.87 ± 0.01	0.055 ± 0.012	0.8
		4 h	50.8 ± 6.7	6.68 ± 0.09	0.048 ± 0.004	0.7
		6 h	50.7 ± 4.1	6.75 ± 0.01	0.087 ± 0.006	1.2
Intestinal	EPDO-40	0 h	441.9 ± 8.6	6.49 ± 0.09	0.197 ± 0.008	3.0
digestion		2 h	435.2 ± 9.4	6.64 ± 0.09	0.211 ± 0.015	3.2
		4 h	441.7 ± 10.5	6.58 ± 0.02	0.208 ± 0.015	3.2
		6 h	438.5 ± 10.8	6.52 ± 0.02	0.213 ± 0.013	3.3
	EPDO-60	0 h	267.5 ± 6.9	7.19 ± 0.11	0.171 ± 0.012	2.4
		2 h	264.6 ± 7.8	7.24 ± 0.06	0.167 ± 0.009	2.3
		4 h	268.9 ± 9.2	7.19 ± 0.08	0.169 ± 0.008	2.4
		6 h	263.8 ± 8.6	7.23 ± 0.04	0.174 ± 0.014	2.4
	EPDO-80	0 h	51.5 ± 4.4	6.78 ± 0.05	0.127 ± 0.012	2.1
		2 h	52.8 ± 2.8	6.85 ± 0.02	0.167 ± 0.006	2.4
		4 h	50.5 ± 5.8	6.83 ± 0.07	0.195 ± 0.011	2.9
		6 h	51.7 ± 3.7	6.76 ± 0.05	0.201 ± 0.008	3.0

Abbreviation: *Mw*: molecular weight; EPDO: ethanol-fractional polysaccharides from *Dendrobium officinale*; ^a^ ND: These data could not be detected; ^b^ /: These data were not calculated.

**Table 2 foods-13-01675-t002:** EPDO variation with time in fermentation.

Sample Name	Fermentation Time (h)	Carbohydrate Content (mg/mL)	Reducing Sugar Content (mg/mL)	Mannose (mg/mL)	Glucose (mg/mL)	Degradation Rate (%)
EPDO-40	0	6.56 ± 0.04	ND ^a^	ND	ND	/ ^b^
	6	5.17 ± 0.06	2.10 ± 0.04	1.28 ± 0.07	0.38 ± 0.06	21.3
	12	4.08 ± 0.11	1.37 ± 0.08	0.59 ± 0.08	0.24 ± 0.04	36.4
	24	3.47 ± 0.17	0.32 ± 0.05	0.13 ± 0.02	0.05 ± 0.01	48.7
EPDO-60	0	7.22 ± 0.08	ND	ND	ND	/
	6	5.11 ± 0.14	2.40 ± 0.15	1.62 ± 0.16	0.33 ± 0.12	28.7
	12	3.89 ± 0.18	1.86 ± 0.22	1.24 ± 0.06	0.23 ± 0.09	47.1
	24	2.61 ± 0.24	1.50 ± 0.20	0.93 ± 0.07	0.18 ± 0.11	63.8
EPDO-80	0	6.88 ± 0.01	ND	ND	ND	/
	6	5.52 ± 0.07	1.98 ± 0.09	1.21 ± 0.07	0.52 ± 0.09	21.0
	12	4.28 ± 0.09	1.30 ± 0.11	0.71 ± 0.03	0.29 ± 0.05	38.2
	24	3.32 ± 0.14	0.46 ± 0.17	0.18 ± 0.04	0.10 ± 0.04	53.1
FOS	0	7.91 ± 0.06	ND	- ^c^	-	/
	6	5.32 ± 0.10	2.66 ± 0.14	-	-	30.8
	12	2.91 ± 0.08	2.01 ± 0.04	-	-	63.2
	24	2.23 ± 0.16	1.52 ± 0.06	-	-	70.4
Blank	0	ND	ND	-	-	/
	6	ND	ND	-	-	/
	12	ND	ND	-	-	/
	24	ND	ND	-	-	/

Abbreviations: EPDO: ethanol-fractional polysaccharides from *Dendrobium officinale*; FOS: fructooligosaccharide; ^a^ ND: These data could not be detected. ^b^ /: These data were not calculated. ^c^ -: These data were not detected.

## Data Availability

The original contributions presented in the study are included in the article/Appendix A, further inquiries can be directed to the corresponding authors.

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
