# Peer review of "In Vitro Digestion and Fermentation of Different Ethanol-Fractional Polysaccharides from Dendrobium officinale: Molecular Decomposition and Regulation on Gut Microbiota"

_foods, 2024, doi:10.3390/foods13111675_

Round 1

Reviewer 1 Report

Comments and Suggestions for Authors

1.      The reasons why the authors conducted this study are not very solid. More beneficial effects on the fermentation of the polysaccharides should be added in the introduction.

2.      Chemical structure changes after the fermentation should be described in the introduction.

3.      The fecal samples from three donors were used in the experiment. Why did three fecal samples be used in this experiment? For human samples, at least 6 biological replicates should be used.

4.      The analysis of gut microbiota profile is comprehensive and in the manuscript, SCFAs is the focus and the KEGG pathways analysis should combine with this.

5.      The manuscript is well designed and managed. And the results are well presented.

6.      For future research, in vivo study is strongly recommended.

Comments on the Quality of English Language

Need to be improved

Author Response

Dear Editor and reviewer,

We sincerely thank the editor and all reviewers for their valuable feedback that we have used to improve the quality of our manuscript. The comments are laid out below in normal font and specific concerns have been numbered. Our response is given in italicized font and changes/additions to the manuscript are given in red. The written English has been revised and checked by Prof. Wu in HKPolyU.

Details of revisions are showed below:

Comment of Reviewer #1

1. The reasons why the authors conducted this study are not very solid. More beneficial effects on the fermentation of the polysaccharides should be added in the introduction.

ANS: Thanks for the comments. Some beneficial effects of polysaccharides on gut flora have been added in the introduction.

 2. Chemical structure changes after the fermentation should be described in the introduction.

ANS: Thanks for the comments. Some examples of chemical structure changes after fermentation have been added in the introduction.

 3. The fecal samples from three donors were used in the experiment. Why did three fecal samples be used in this experiment? For human samples, at least 6 biological replicates should be used.

ANS: Thanks for the comments. We conducted six parallel experiments, but due to experimental anomalies, some data was lost. Consequently, when collecting the data, we only retained the three most complete datasets.

4. The analysis of gut microbiota profile is comprehensive in the manuscript, SCFAs is the focus and the KEGG pathways analysis should combine with this.

ANS: Thanks for the comments. We combined SCFA with KEGG path analysis appropriately and have add it to section 3.5 and section 4.5.

5. The manuscript is well designed and managed. And the results are well presented.

ANS: Thanks for your kindly comments.

 6. For future research, in vivo study is strongly recommended.

ANS: Thanks for the comments. We are currently designing in vivo experiments with the goal of developing superior health foods for the future.  Furthermore, in our conclusion, we have outlined the potential application of polysaccharides in the development of innovative foods.

We tried our best to improve the manuscript and made some changes to the manuscript. These changes will not influence the content and framework of the paper. And here we did not list the changes but marked in red in the revised paper. We appreciate for Editors warm work earnestly and hope that the correction will meet with approval.

Reviewer 2 Report

Comments and Suggestions for Authors

This study investigated the digestive properties and regulation effects of ethanol-fractional polysaccharides from Dendrobium officinale on gut microbiota in vitro. Following simulated digestion of saliva, gastric, and small intestine phases, all three EPDOs exhibited remarkable stability as they entered the large intestine. In vitro fermentation revealed a "melting" decomposition pattern of EPDOs. Moreover, EPDO-60 significantly increased the relative abundance of Bacteroides, a key genus involved in polysaccharide metabolism. Additionally, EPDO-60 promoted the growth of specific microbiota, indicating potential health benefits associated with these polysaccharides. These findings highlight the potential of EPDO-60, in promoting gut health through their digestive properties and regulation of gut microbiota.

I find the document very interesting, and from my point of view, it has enough elements for publication. I will request some corrections to improve the manuscript:

202 Change Chaol to Chao1

226 It is recommended that the results of your research be archived in the NCBI database

252 Table 1, 279 Table 2 Remember that you must define all abbreviations and briefly describe what you are showing, we must ensure that the tables and figures can be understood by the reader, even without reading the entire manuscript.

Figures 7 and 9 are not well-dimensioned, in my opinion, the results of the analysis presented in these figures are a core part of the experiment. It is not possible to read or appreciate what they show.

I don't understand Fig 10, it doesn't have any description

Conclusions. I suggest that in this section the authors describe the potential applications it would have in the innovation and development of new foods.

Author Response

Dear Editor and reviewer,

We sincerely thank the editor and all reviewers for their valuable feedback that we have used to improve the quality of our manuscript. The comments are laid out below in normal font and specific concerns have been numbered. Our response is given in italicized font and changes/additions to the manuscript are given in red. The written English has been revised and checked by Prof. Wu in HKPolyU.

Details of revisions are showed below:

Comment of Reviewer #2:

1. 202 Change Chaol to Chao1

ANS: Thanks for the comments. We are very sorry for our careless and it was rectified in Line 213.

 2. 226 It is recommended that the results of your research be archived in the NCBI database

ANS: Thanks for the comments. The results of our research have been archived in the NCBI database, and the revision has been added in line 237.

3. 252 Table 1, 279 Table 2 Remember that you must define all abbreviations and briefly describe what you are showing, we must ensure that the tables and figures can be understood by the reader, even without reading the entire manuscript.

ANS: Thanks for the comments. The abbreviations have been added in lines 264 - 291.

 4. Figures 7 and 9 are not well-dimensioned, in my opinion, the results of the analysis presented in these figures are a core part of the experiment. It is not possible to read or appreciate what they show.

ANS: Thanks for the comments. We have re-scaled Figures 7 and 9.

 5. I don't understand Fig 10, it doesn't have any description

ANS: Thanks for the comments. We are very sorry for our careless in placing Figure 10 incorrectly, which caused confusion for you. Figure 10 serves as a summary for the entire text, and we have now included its description in the discussion section.

 6. Conclusions. I suggest that in this section the authors describe the potential applications it would have in the innovation and development of new foods.

ANS: Thanks for the comments. The potential applications of new foods have been added to discussion.

We tried our best to improve the manuscript and made some changes to the manuscript. These changes will not influence the content and framework of the paper. And here we did not list the changes but marked in red in the revised paper. We appreciate for Editors warm work earnestly and hope that the correction will meet with approval.